# Effect of Computer-Aided Navigation Techniques on the Accuracy of Endodontic Access Cavities: A Systematic Review and Meta-Analysis

**DOI:** 10.3390/biology10030212

**Published:** 2021-03-10

**Authors:** Álvaro Zubizarreta-Macho, Sara Valle Castaño, José María Montiel-Company, Jesús Mena-Álvarez

**Affiliations:** 1Department of Endodontics, Faculty of Health Sciences, Alfonso X El Sabio University, 28691 Madrid, Spain; saravalle@hotmail.es (S.V.C.); Jmenaalv@uax.es (J.M.-Á.); 2Department of Stomatology, Faculty of Medicine and Dentistry, University of Valencia, 46010 Valencia, Spain; jose.maria.montiel@uv.es

**Keywords:** computer-assisted treatment, endodontic access cavity, endodontics, image-guided treatment, navigation system, real-time tracking

## Abstract

**Simple Summary:**

The endodontic access cavity is an essential step in the root canal treatment allowing the access and location of the root canal system. The inaccuracy during the endodontic access cavitie can lead to intraoperative complications such as missed root canals or root perforations, and hence affect the prognosis of the root canal treatment. The development of computer-aided static and dynamic navigation techniques has improved the accuracy of endodontic access cavities and root canal location; however, it is necessary a comparative analysis to clarify which of the computer-aided navigation system is more accurate. This systematic review and meta-analysis aims to clarify the root canal success rate using static and/or dynamic navigation systems compared to freehand accesses.

**Abstract:**

The present systematic review and meta-analysis aims to determine the effect of computer-aided navigation techniques on the accuracy of endodontic access cavities. Materials and methods: A systematic literature review and meta-analysis of clinical studies, based on Preferred Reporting Items for Systematic Reviews and Meta-Analyses (PRISMA) recommendations, was performed that evaluated the root canal location rate of computer-aided navigation techniques applied to endodontic access cavities. Four different databases were used to consult the literature: PubMed-Medline, Scopus, Cochrane, and Web of Science. After discarding duplicate articles and applying inclusion criteria, 14 articles were selected for qualitative analysis and 13 for quantitative analysis. Results: the root canal location success rate started at 98.1% (CI: 95.7–100%) of the cases performed through a computer-aided navigation technique. The prediction interval ranged from 93.3% to 100%. The meta-analysis did not detect heterogeneity between the combined studies (Q-test = 17.3; *p* = 0.185; I^2^ = 25%). No statistically significant differences were found between computer-aided static navigation techniques (success rate: 98.5%) and computer-aided dynamic navigation techniques (success rate: 94.5%) (Q test = 0.57; *p* = 0.451), nor between in vitro studies (success rate: 96.2%) and in vivo studies (success rate: 100%) (Q test = 2.53; *p*-value = 0.112). An odds success ratio of 13.1 (CI: 95%; 3.48, 49.1) encourages the use of computer-aided navigation techniques over conventional endodontic access cavity procedures. Conclusions: the endodontic access cavities created using static and dynamic computer-aided navigation techniques are highly accurate in locating the root canal system.

## 1. Introduction

Locating the root canal system is such an essential procedure in root canal treatment, as missed root canals can affect the prognosis of the root canal treatment and therefore the survival of the tooth [1]. Karabucak et al. reported an incidence of 23% of missed root canals through a Cone Beam Computed Tomography (CBCT)-based study of a North American population, with a 4.38-fold higher risk of developing apical periodontitis linked to the omitted root canals [2]. The endodontic access cavities must allow for complete location of the root canal system and direct access of the endodontic instruments to the root canal system, facilitate disinfection and complete debridement, and help avoid excessive loss of the dental structure [3]. Improvements on radiodiagnostic test such as Cone Beam Computed Tomography (CBCT) has enabled better knowledge of root canal system location and distribution, improving the root canal treatment success rate [4]. In addition, the development of computer-aided static (SN) and dynamic (DN) navigation techniques has helped to guide drilling during endodontic access cavity procedures [5]. Both computer-aided navigation techniques are based on CBCT datasets; however, computer-aided SN techniques require computer-aided design and computer-aided manufacturing of surgical templates using rapid prototyping techniques, while computer-aided DN techniques require an optical triangulation tracking system that uses stereoscopic motion-tracking cameras to guide the drilling process at the planned angle, pathway, and depth of the endodontic access cavities in real time. Computer-aided navigation techniques enable more accurate and safer endodontic access to cavities than conventional freehand techniques [6]. Inaccurate endodontic access cavities may lead to intraoperative complications such as overextended access cavities, crown perforation, root perforation, missed root canals, fracture of root canal instruments during canal preparation [3], or weakening of the coronal structure [6].

The present systematic review and meta-analysis aims to analyze the effects of computer-aided navigation techniques on the accuracy of endodontic access cavities assessed via a systematic review and meta-analysis with a null hypothesis (H0) stating that there is no difference between the effects of different computer-aided navigation techniques on the accuracy of endodontic access cavities.

## 2. Materials and Methods

### Study Design

The literature review was conducted following guidelines for systematic reviews and meta-analyses in accordance with PRISMA (Preferred Reporting Items for Systemic Reviews and Meta-Analyses http://www.prisma-statement.org (accessed on 4 May 2020); International Prospective Register of Systematic Reviews (PROSPERO) registration number: CRD42020192179). The review also complied with the PRISMA 2009 Checklist [7] and was performed in accordance with current recommendations with regard to endodontic systematic reviews and meta-analyses [8,9]. The population, intervention, comparison, and outcome (PICO) question was “What is the effect of computer-aided navigation techniques on the accuracy of endodontic access cavities?”, with the following components:

Population: endodontic access cavities performed in teeth using computer-aided navigation techniques;

Intervention: endodontic access cavities performed using computer-aided navigation techniques;

Comparison: endodontic access cavities performed using static (SN) or dynamic navigation (DN) systems; and

Outcome: accuracy and canal location of endodontic access cavities.

An electronic search was carried out using the following databases: PubMed, Scopus, Cochrane, and Web of Sciences. The search assessed all the literature published internationally through to April 2020. Seven medical subject heading (MeSH) terms were included in the search: “endodontic access cavity”; “conservative access cavity”; “guided access cavity”; “navigation access cavity”; “ninja access cavity”; “accuracy”; “deviation”; “canal location”; and “dental implants”. Three Boolean operators were applied (“OR”, “AND”, and “NOT”). These search terms were applied as follows: [(“endodontic access cavity”) OR (“conservative access cavity”) OR (“guided access cavity”) OR (“navigation access cavity”) OR (“ninja access cavity”)] AND [(“accuracy”) OR (“deviation”) OR (“canal location”)] NOT [(“dental implants”)]. Two different researchers (S.V.C.; A.Z.M.) searched the databases simultaneously. The inclusion and exclusion criteria were applied to titles, and a single researcher (S.V.C.) extracted the data regarding the relevant variables. A.Z.M. conducted the systematic review, and two researchers who had not participated in the selection process (A.Z.M.; J.M.M.C.) performed the subsequent meta-analysis.

The inclusion criteria were as follows: randomized experimental trials (RETs), clinical trials, and case series (CS) studies of 2 patients were included in the database. Teeth in which endodontic access cavities were performed using computer-aided techniques (either static or dynamic navigation techniques) were included. Studies were not restricted by language or year of publication. The exclusion criteria were as follows: systematic literature reviews, prospective and retrospective randomized clinical trials, clinical cases, and editorials. Moreover, studies that did not provide information related to the root canal location or presented a sample size smaller than the stablished were rejected. The following data were recorded: author, year, title, journal, sample size (n), and accuracy of canal location. The results obtained from studies that analyzed the accuracy of the endodontic access cavities using static and/or dynamic navigation systems were included.

The Current Research Information System (CRIS) scale was used to assess the methodological quality of the selected in vitro studies, which is composed of four items that analyze the sample preparation, the randomization and blinding procedures and the statistical test. The best-rated studies were those that met all the concepts; if 2–3 variables were present, they were rated as fair quality; and studies in which no or only one aspect was covered were classed as poor quality [10]. The Jadad was used to assess the methodological quality of the selected in clinical studies to evaluate risk of bias. This scale comprises five items that assess randomization, researcher and patient blinding, and a description of losses during follow-up, resulting in a final score of 0–5, with scores less than 3 being deemed low quality [11].

The meta-analysis was carried out using a random effects model with the inverse of the variance method and the Mantel–Haenszel method. The estimated effect size was analyzed using the root canal location success ratio and the odds ratio with a 95% confidence interval. The existence of heterogeneity between the combined studies was assessed using the Q test (*p* < 0.05) and quantified with the I^2^ statistical index proposed by Higgins, which describes the percentage of the total variation between the studies due to heterogeneity rather than being random, quantifying the effect of heterogeneity between 0 and 100%; 25–50% was considered mild, 50–75% was considered moderate, and >75% was considered high. The Q intergroups test (*p* < 0.05) was used to assess the existence of differences in the success rate between the subgroups. Meta-analyses were represented by forest plots. Publication bias was assessed by comparing the initial estimated success rate with the adjusted rate using the Trim and Fill method, plotted with Funnel plots.

## 3. Results

### 3.1. Flow Diagram

32 articles in PubMed, 4 articles in Web of Sciences, 164 articles in Cochrane and 78 articles in Scopus were found after the initial search. Of these 278 works, 16 duplicates were discarded. After assessing study titles and abstracts, another 176 were eliminated, after which 86 remained. An additional 61 were rejected for failing to fulfill the inclusion criteria by either not including the canal location rate or not meeting the minimum sample size. After this selection process, a total of 14 articles was selected for final qualitative synthesis. Thirteen articles were ultimately assessed in the quantitative analysis, as they met all the selection criteria (Figure 1).

### 3.2. Qualitative Analysis 

Of the 13 articles included, 8 were experimental trials [4,6,12,13,14,15,16,17], 4 were CS [18,19,20,21], and 1 was a clinical trial (CT) [22]. In addition, one study compared static navigation versus conventional endodontic access cavities [16], and one study compared static and dynamic navigation versus conventional endodontic access [6]. Most studies showedd a sample size greater than 10, even though they ranged from as low as 2 in the study by Hu Chen (2018) [17], to as high as 60 in Connert’s study in 2017 [15]. However, most of the clinical studies were CS with 2–3 patients [19,20,21,22], and only one CT was included with a sample size of 50 patients [23] (Table 1).

### 3.3. Quality Assessment

Table 2 shows the results of the methodological quality assessment using the CRIS scale. One article [14] showed 1 point at the CRIS scale, resulting poor methodological quality, six articles [4,15,16,17,18,19] obtained a score of 2, and one article [6] obtained a score of 3 on the CRIS scale, indicating its high methodological quality. Quality scores were most often compromised by failure to fulfill criteria related to the randomization and blinding process.

Table 3 shows the results obtained atthe Jadad scale. The Jadad scale determined four articles to be “not applicable” because they were part of a case series [19,20,21,22], and the authors of these articles neither blinded nor randomized the studies. The only CT [23] obtained a score of 0, indicating poor methodological quality, which was compromised by failure to comply with items regarding the randomization and blinding process.

### 3.4. Quantitative Analysis

#### 3.4.1. Root Canal Location Success Rate

Fourteen results from thirteen studies were selected and combined using a random effects model with an inverse variance method. The root canal location success rate was stablished at 98.1% (CI: 95.7–100%), and all were performed using a computer-aided technique (Figure 2). The prediction interval ranged from 93.3% to 100%. The meta-analysis did not detect heterogeneity between the combined studies (Q-test = 17.3; *p* = 0.185; I^2^ = 25%) (Figure 2).

The subgroup analysis did not detect any statistically significant differences between the root canal location success rate of computer-aided navigation techniques (Q test = 0.57; *p* = 0.451) (Figure 3). The SN computer-aided technique (12 studies) showed a root canal location success rate of 98.5% with a confidence interval between 96.1% and 100%; in comparison, the DN computer-aided techniques (two studies) showed a root canal location success rate of 94.5%, with a confidence interval between 84.4% and 100% (Figure 3).

There were also no statistically significant differences related to the study type (Q test = 2.53; *p* = 0.112) between in vitro studies (nine studies), with 96.2% and a confidence interval between 92.4% and 100%, and in vivo studies (five studies), with 100% and a confidence interval between 97.3% and 100% (Figure 4).

#### 3.4.2. Comparison Between Computer-Aided Navigation Techniques and Control Group

Three results from two studies using computer-aided navigation techniques and control group were analyzed using a random effects model with the Mantel–Haenszel method. No heterogeneity was observed (Q test = 0.10; *p* = 0.949; I^2^ = 0%). An Odds success ratio of 13.1 (CI: 95%; 3.48, 49.1) favoring the use of a navigation system was estimated (Figure 5).

### 3.5. Publication Bias

Using the Trim and Fill method, no studies were added to obtain a symmetrical image of the funnel plot. The estimate of the success rate did not change (98.1%; CI: 95% between 95.7% and 100%), and the two funnel plots (initial and adjusted) showed identical images, indicating an absence of publication bias (Figure 6).

## 4. Discussion

The results obtained in the present study confirm the null hypothesis (H0), which holds that there is no difference between the effects of computer-aided navigation techniques on the accuracy of endodontic access cavities.

Many authors have evaluated the accuracy of computer-aided static navigation techniques for dental implant placement. These authors showed a 0.99 mm horizontal deviation (ranging from 0.0 mm to 6.5 mm) at the dental implant platform, a 1.24 mm horizontal deviation (ranging from 0.0 mm to 6.9 mm) at the dental implant apex, and an average angle deviation of 3.81° (ranging from 0.0° to 24.0°) relative to the longitudinal axis of dental implants [24,25]. However, Gambarini et al. reported a mean horizontal deviation of 0.34 mm and a mean angle deviation of 4.8° using a dynamic navigation system to perform ultraconservative access cavities [26]. This inaccuracy has lent credibility to the potential application of computer-aided dynamic navigation techniques to clinically transfer the positions of virtually planned dental implants [27], showing lower deviation values at the dental implant platform (0.71 ± 0.40 mm) and at the dental implant apex (1.00 ± 0.49 mm), and an angle deviation (2.26° ± 1.62°) relative to the longitudinal axis of the dental implants [28]. These techniques have resulted in greater accuracy of the freehand dental implant placement technique and reduced clinical complications, which enable the technique to be safer and more predictable [29,30]. These small deviations are not clinically relevant for implant surgery; however, it is important that computer-aided navigation techniques used in endodontics be accurate, because root canal location requires a high level of accuracy, especially in calcific metamorphosis [20,31,32,33] or dental malformations such as dens invaginatus [34,35] or dens evaginatus [36]. In addition, computer-assisted static navigation techniques have also been used in endodontic surgery to perform the alveolar processes of autotransplanted teeth [37], as well as to ensure accurate root-end resection in endodontic microsurgery [38,39]. However, smaller or limited mouth openings or posterior teeth with limited access prevent the use of computer-assisted static navigation techniques [40]. Furthermore, design and/or manufacturing errors in the endodontic template may lead to intraoperative complications that cannot be solved during the endodontic access cavity. However, computer-aided dynamic navigation techniques enable a direct view into the endodontic access cavity and enable clinicians to readjust the direction of the endodontic access cavity bur [6,14,24]. The primary disadvantage of computer-aided dynamic navigation techniques is the difficulty in maintaining visibility of the system display when creating the endodontic access cavity, as well as the long learning curve required [6,14,24]. However, augmented reality devices can reportedly be used to transfer over the virtual image displayed by the computer-aided dynamic navigation system while maintaining visibility of the therapeutic field [41]. In addition, virtual reality has been used to perform endodontic access cavities [42].

Locating the root canal using computer-assisted navigation techniques has had a high success rate (98.1%) without statistically significant differences (*p* = 0.451) between static or dynamic computer-assisted navigation techniques. That being said, computer-assisted static navigation techniques showed a slightly higher root canal location success rate (98.5%) than computer-assisted dynamic navigation techniques (94.5%). These results may be influenced by the small number of studies on the dynamic technique of computer-assisted navigation (two studies) with respect to the larger number of studies on the technique of static-assisted computer navigation (12 studies). Furthermore, the computer-aided static navigation technique emerged before the computer-aided dynamic navigation technique; therefore, the computer-aided static navigation technique might have been further improved and documented in the scientific literature. In any case, regardless of the computer-assisted navigation technique analyzed, both procedures have improved the success rate of locating the root canal attributed to conventional endodontic access cavities; therefore, computer-assisted static and dynamic navigation techniques can be recommended due to their high efficacy rates in root canal location, especially in cases of calcific metamorphosis or dental malformations. In vivo studies have shown a notably higher success rate of the root canal location (100%) compared with in vitro studies (96.2%). One would expect precisely the opposite, because in vitro studies allow for the possibility of controlling clinical variables that could influence the results of the studies. However, the higher success rate attributed to in vivo studies confirms the efficacy and safety of computer-aided navigation techniques.

This studypresent as limitation the risk of not finding related articles, although this risk was decreased given that four databases were searched. Clinical studies were of poor quality, with a score of zero as per the Jadad scale criteria. However, most in vitro studies were of high quality, with scores between two and three as per the CRIS criteria. Furthermore, a few randomized studies, both clinical and in vitro, were also included. Only two studies of DN computer-aided techniques and two clinical studies were included, so further clinical studies of higher quality and better design are needed to corroborate the results.

## 5. Conclusions

The conclusion derived from the present study is that static and dynamic computer-aided navigation techniques are highly accurate in locating root canal systems in order to perform endodontic access cavities.

## Figures and Tables

**Figure 1 biology-10-00212-f001:**
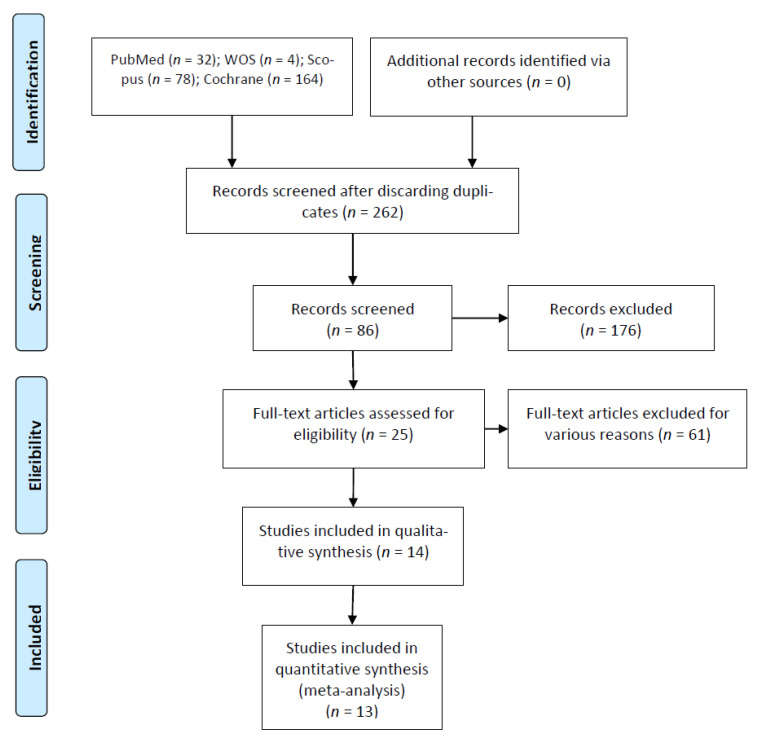
Preferred Reporting Items for Systematic Reviews and Meta-Analyses (PRISMA) Flow diagram.

**Figure 2 biology-10-00212-f002:**
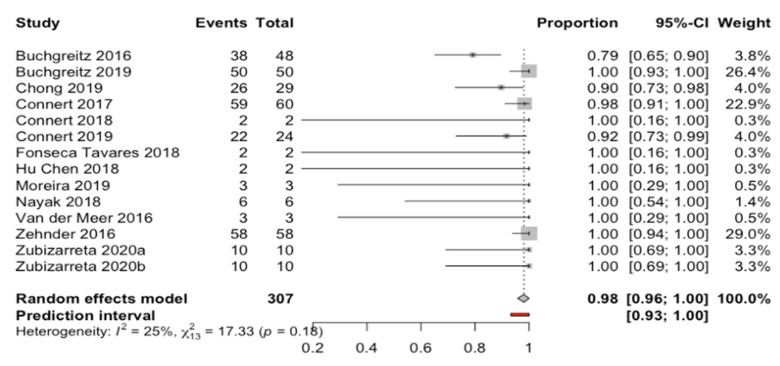
Forest plot of root canal location success rate between the studies selected. The horizontal axis expresses the root canal proportion success rate of each study.

**Figure 3 biology-10-00212-f003:**
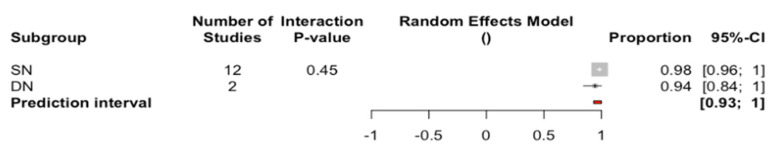
Forest plot of root canal location success rate compared between computer-aided navigation techniques. The horizontal axis expresses the root canal proportion success rate of each computer-aided navigation technique.

**Figure 4 biology-10-00212-f004:**
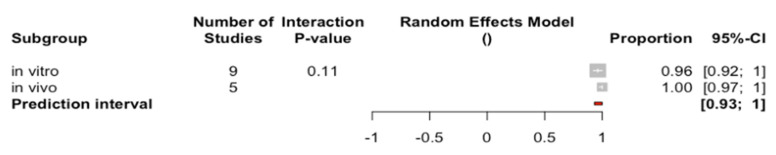
Forest plot of root canal location success rate between study types. The horizontal axis expresses the root canal proportion Scheme.

**Figure 5 biology-10-00212-f005:**
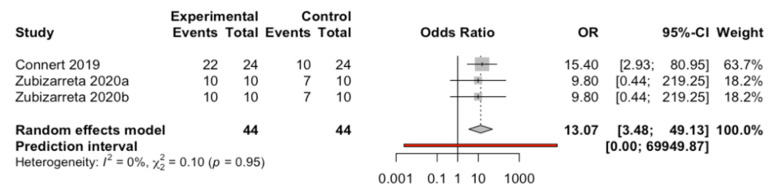
Forest plot of root canal location success OR between computer-aided navigation techniques and control group. The horizontal axis expresses the root canal proportion success rate of each study.

**Figure 6 biology-10-00212-f006:**
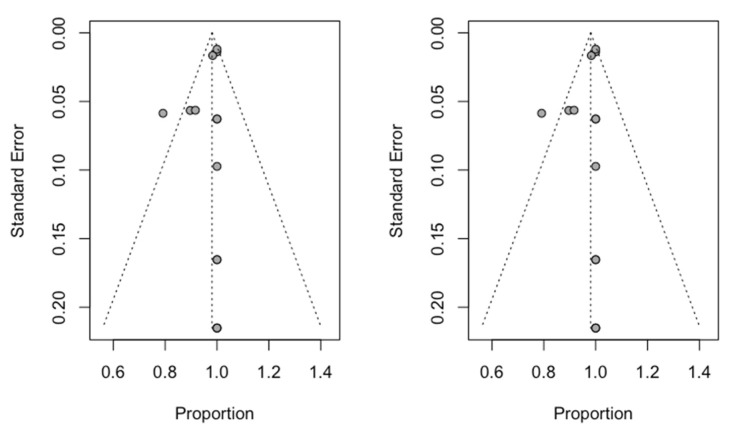
Initial Funnel plot and after Trim and Fill adjustment.

**Table 1 biology-10-00212-t001:** Qualitative analysis of articles forming part of the systematic review.

Author (Year)	Study Type	Sample (*n*)	Measurement Procedure	Computer-Aided Navigation Technique	Root Canal Location Rate	Computer-Aided Navigation Technique Results
Buchgreitz et al. (2016) [13]	In vitro	48	Clinical and radiographic	SN	38/48	Apical horizontal deviation: 0.46 (0.69–0.32) mm
Buchgreitz et al. (2019) [23]	CT	50	Clinical and radiographic	SN	50/50	Root canal location success: 100%
Chong et al. (2019) [14]	In vitro	29	Clinical and radiographic	DN	26/29	Root canal location success: 89.7%
Connert et al. (2017) [15]	In vitro	60	Clinical and radiographic	SN	59/60	Base of the bur (angle) deviation: 1.59 ± 1.22°
						Base of the bur (mesio–distal) deviation: 0.12 ± 0.12 mm
						Base of the bur (buccal–oral) deviation: 0.13 ± 0.13 mm
						Base of the bur (apical–coronal) deviation: 0.12 ± 0.12 mm
						Tip of the bur (mesio–distal) deviation: 0.14 ± 0.18 mm
						Tip of the bur (buccal–oral) deviation: 0.34 ± 0.28 mm
						Tip of the bur (apical–coronal) deviation: 0.12 ± 0.11 mm
Connert et al. (2018) [19]	CS	2	Clinical and radiographic	SN	2/2	Root canal location success: 100%
Connert et al. (2019) [16]	In vitro	48	Clinical and radiographic	Control	10/24	Root canal location success: 41.7%
						Substance loss: 49.9 mm^3^
				SN	22/24	Root canal location success: 91.7%
						Substance loss: 9.8 mm^3^
Fonseca Tavares et al. (2018) [20]	CS	2	Clinical and radiographic	SN	2/2	Root canal location success: 100%
Hu Chen et al. (2018) [17]	In vitro	2	Clinical and radiographic	SN	2/2	Root canal location success: 100%
Jain et al. (2020) [23]	In vitro	138	Clinical and radiographic	DN	NAv	Apical horizontal deviation 2D: 0.9
						Apical horizontal deviation 3D: 1.3
						Angular deviation 3D: 1.7
Moreira Maia et al. (2019) [21]	CS	3	Clinical and radiographic	SN	3/3	Root canal location success: 100%
Nayak et al. (2018) [18]	In vitro	6	Clinical and radiographic	SN	6/6	Buccal–lingual deviation: 0.07–0.20 mm
						Mesio–distal deviation: 0.08–0.19 mm
						Total deviation: 0.15–0.26 mm
Van der Meer et al. (2016 [22]	CS	3	Clinical and radiographic	SN	3/3	Root canal location success: 100%
Zhender et al. (2016) [4]	In vitro	58	Clinical and radiographic	SN	58/58	Angle deviation: 1.81°
						Mesio–distal deviation: 0.21 mm
						Buccal–palatal deviation: 0.2 mm
						Apical–coronal deviation: 0.16 mm
Zubizarreta et al. (2020) [6]	In vitro	30	Clinical and radiographic	Control	7/10	Coronal deviation: 4.03 ± 1.93 mm
						Apical deviation: 2.43 ± 1.23 mm
						Angular deviation: 14.95 ± 11.15°
				a: SN	10/10	Coronal deviation: 7.44 ± 1.57 mm
						Apical deviation: 7.13 ± 1.73 mm
						Angular deviation: 10.04 ± 5.2°
				b: DN	10/10	Coronal deviation: 3.14 ± 0.86 mm
						Apical deviation: 2.48 ± 0.94 mm
						Angular deviation: 5.58 ± 3.23°

RCT: Randomized Controlled Trial; CT: Controlled Trial; CS: Case Series; NAv: Not Available; PL: platelet enriched plasma; Os: bone graft; MB: membrane; SN: Static Navigation; DN: Dynamic Navigation.

**Table 2 biology-10-00212-t002:** Methodological quality assessment as per the Current Research Information System (CRIS) scale.

Author (Year)	Sample Preparation and Handling	Allocation Sequence and RanDomization Process	Whether the Evaluators Were Blinded	Statistical Analysis	Score
Buchgreitz et al. (2016) [13]	Yes	No	No	Yes	2
Chong et al. (2019) [14]	Yes	No	No	No	1
Connert et al. (2017) [15]	Yes	No	No	Yes	2
Connert et al. (2019) [16]	Yes	No	No	Yes	2
Hu Chen et al. (2018) [17]	Yes	No	No	Yes	2
Nayak et al. (2017) [18]	Yes	No	No	Yes	2
Zehnder et al. (2017) [4]	Yes	No	No	Yes	2
Zubizarreta et al. (2020) [6]	Yes	Yes	No	Yes	3

**Table 3 biology-10-00212-t003:** Methodological quality assessment using the Jadad scale.

Jadad Criteria						
Author (Year)	Is the Study Randomized?	Is the Study Double-Blinded?	Were Withdrawals and Dropouts Described?	Adequate Method of Randomization?	Appropriate Blinding Method?	Score
Buchgreitz et al. (2019) [23]	0	0	0	0	0	0
Connert et al. (2018) [19]	NA	NA	NA	NA	NA	NA
Fonseca Tavares et al. (2018) [20]	NA	NA	NA	NA	NA	NA
Moreira Maia et al. (2019) [21]	NA	NA	NA	NA	NA	NA
Van der Meer et al. (2016) [22]	NA	NA	NA	NA	NA	NA

NA: Not applicable.

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
