# Peer review of "Effect of Computer-Aided Navigation Techniques on the Accuracy of Endodontic Access Cavities: A Systematic Review and Meta-Analysis"

_biology, 2021, doi:10.3390/biology10030212_

Round 1

Reviewer 1 Report

The paper is a very interesting paper. Considering the methodology the paper has great quality. The discussion could be improved according to the results obtained. The english shoul be improved.

Author Response

Dear Reviewer 1:

I’m pleased to resubmit the manuscript of the work entitled, “Efficacy of Computer-Aided Navigation Techniques on the Accuracy of
Endodontic Access Cavities. A Systematic Review and Meta-Analysis”

Reviewer 1: English language and style are fine/minor spell check required

Response: In order to adapt to the reviewer's 1 comments, we have send the manuscript to the English Editing Service of MDPI. We attached the Certificate.

We take this opportunity to thank the recommendations and suggestions made by the reviewers to improve the document.

Yours sincerely,

Reviewer 2 Report

The paper is a systematic review and meta-analysis on the effect of computer-aided navigation techniques on the accuracy of endodontic access cavities.
The authors made a great work in terms of methodology and the paper sounds scientific and well written.

However some improvements are mandatory before acceptance.

In Abstract section, kindly write Keywords in alphabetical order.

In Introduction:

Pag 1 lines 13-15: Kindly check these sentences. The main clause seems to miss: “A

systematic literature review and meta-analysis of clinical studies that evaluate the root

canal location rate of computer-aided navigation techniques as applied to endodontic

access cavities, based on Preferred Reporting Items for Systematic Reviews and Meta-

Analyses (PRISMA) recommendations”.

Page 1 lines 34-35: Authors should check reference [1] since it discusses about a dens invaginatus case report and it seems quite different from what stated with the following sentences: “Locating the root canal system is an essential procedure in root canal treatment, to such an extent that missed root canals can affect the prognosis of the root canal treatment and therefore the survival of the tooth [1]”. A reference that really evaluated the relationship between missed root canals and prognosis of root canal treatment could be more appropriate, in order to scientifically support the above-mentioned comment.

Materials and methods are clear and well explained. Different aspects are analyzed with dedicated statistical tests. The authors did a great job in the explication of all the variables identified and
included in the study.

Nevertheless, some improvements are mandatory in Materials and Methods:

  • Page 2 lines 77-78: Kindly clarify the databases used in this research, since in Materials and Methods section “Cochrane” was not included, while in the Result section “Cochrane” was mentioned (Pag 3 line 130). “an electronic search was carried out using the following databases: PubMed, Scopus, Embase, and Web of Sciences”. On the contrary, “Embase” was mentioned in Materials and Methods and was not mentioned in Results. Kindly specify if searching on Embase no suitable article was found.
  • Page 2 line 80-86: Please provide the exact search string used in the different databases.
  • Page 2 line 92-94 Please explain the reasons that led you to this choice. Why have you established this inclusion criteria?
  • Page 3 line 118: kindly correct this sentence (lapsus calami)
    I2statistical index” -> “I2 statistical index
  • Page 3 line 121: kindly correct this sentence (lapsus calami)
    <“…0 and 100%.25–50% was” -> “0 and 100%. 25–50% was

Results are easy to understand and comprehensive. All the studied characteristics were reported in tables which are very clear.

Nevertheless, some improvements are mandatory in Results:

  • Page 3 line 130: Kindly check the spelling “…16duplicates…” -> “16 duplicates
  • Page 3 line 132-133 “An additional 61 were rejected for failing to fulfil the inclusion criteria: either not including the canal location rate or not meeting the minimum sample size”. These 61 articles have been deleted because they don’t meet the inclusion criteria, which are indicated in this section more specifically than in the materials and methods section. Please enrich the description in Materials and Methods section.
  • Page 4 line 141 “… and one was a clinical trial (CT) [22]” also clinical trial (CT) has passed the inclusion criteria? Please explain in a more discursive way.
  • Pag 4 line 143: Kindly add the reference of the mentioned study “and one study compared static and dynamic navigation with regard to conventional endodontic access”.

Discussion: this section is complete and evaluates the outcome of different papers present in literature. The overall is comprehensive, concise and complete in its various aspects.
I suggest the following articles to enrich the introduction about this aspect:

“Gambarini G, Galli M, Morese A, Stefanelli LV, Abduljabbar F, Giovarruscio M, Di Nardo D, Seracchiani M, Testarelli L. Precision of Dynamic Navigation to Perform Endodontic Ultraconservative Access Cavities: A Preliminary In Vitro Analysis. J Endod. 2020 Sep;46(9):1286-1290. doi: 10.1016/j.joen.2020.05.022. Epub 2020 Jun 15. PMID: 32553875.”

Please check:

  • Page 9 line 225: kindly check the spelling “deviation (2.26° ± 1.62°)relative to” -> “deviation (2.26° ± 1.62°) relative to
  • Page 9 lines 231-232: I suggest adding reference [1] at this point. “dental malformations such as dens invaginatus [33,34] or dens evaginatus [35].

Conclusions are concise and clear.

Bibliography is not formatted respecting the journal’s requirements, no improper citations are evidenced. Bibliography is written in ascending order.

Figures and labels are clear and easy to comprehend.

English is clear and easy to understand.

Author Response

Dear Reviewer 2:

I’m pleased to resubmit the manuscript of the work entitled, “Efficacy of Computer-Aided Navigation Techniques on the Accuracy of Endodontic Access Cavities. A Systematic Review and Meta-Analysis”

Reviewer 2: English language and style are fine/minor spell check required

Response: In order to adapt to the reviewer's 2 comments, we have send the manuscript to the English Editing Service of MDPI. We attached the Certificate.

Reviewer 2: In Abstract section, kindly write Keywords in alphabetical order.

Response: In order to adapt to the reviewer's 2 comments, we have written the keywords in alphabetical order.

Reviewer 2: In Abstract section, Pag 1 lines 13-15: Kindly check these sentences. The main clause seems to miss: “A systematic literature review and meta-analysis of clinical studies that evaluate the root canal location rate of computer-aided navigation techniques as applied to endodontic access cavities, based on Preferred Reporting Items for Systematic Reviews and Meta-Analyses (PRISMA) recommendations”.

Response: In order to adapt to the reviewer's 2 comments, we have corrected the sentence.

Reviewer 2: In Introduction section, Page 1 lines 34-35: Authors should check reference [1] since it discusses about a dens invaginatus case report and it seems quite different from what stated with the following sentences: “Locating the root canal system is an essential procedure in root canal treatment, to such an extent that missed root canals can affect the prognosis of the root canal treatment and therefore the survival of the tooth [1]”. A reference that really evaluated the relationship between missed root canals and prognosis of root canal treatment could be more appropriate, in order to scientifically support the above-mentioned comment.

Response: In order to adapt to the reviewer's 2 comments, we have changed the reference.

Reviewer 2: In Material and Methods section, Page 2 lines 77-78: Kindly clarify the databases used in this research, since in Materials and Methods section “Cochrane” was not included, while in the Result section “Cochrane” was mentioned (Pag 3 line 130). “an electronic search was carried out using the following databases: PubMed, Scopus, Embase, and Web of Sciences”. On the contrary, “Embase” was mentioned in Materials and Methods and was not mentioned in Results. Kindly specify if searching on Embase no suitable article was found.

Reviewer 2: In Material and Methods section, Page 2 line 80-86: Please provide the exact search string used in the different databases.

Response: In order to adapt to the reviewer's 2 comments, we clarify that we have used the same search string in all the selected databases.

Reviewer 2: In Material and Methods section, Page 2 line 92-94 Please explain the reasons that led you to this choice. Why have you established this inclusion criteria?

Response: In order to adapt to the reviewer's 2 comments, we clarify that we have selected this inclusion criteria to include most of the data present in the scientific literature, since there is not much related literature.

Reviewer 2: In Material and Methods section, Page 3 line 118: kindly correct this sentence (lapsus calami) “I2statistical index” -> “I2 statistical index”

Reviewer 2: In Material and Methods section, Page 3 line 121: kindly correct this sentence (lapsus calami) <“…0 and 100%.25–50% was” -> “0 and 100%. 25–50% was”

Response: In order to adapt to the reviewer's 2 comments, we have corrected this sentence.

Reviewer 2: In Results section, Page 3 line 130: Kindly check the spelling “…16duplicates…” -> “16 duplicates”

Response: In order to adapt to the reviewer's 2 comments, we have corrected this sentence.

Reviewer 2: In Results section, Page 3 line 132-133 “An additional 61 were rejected for failing to fulfil the inclusion criteria: either not including the canal location rate or not meeting the minimum sample size”. These 61 articles have been deleted because they don’t meet the inclusion criteria, which are indicated in this section more specifically than in the materials and methods section. Please enrich the description in Materials and Methods section.

Response: In order to adapt to the reviewer's 2 comments, we have included the exclusion criteria in the Material and Methods section in order to enrich this section.

Reviewer 2: In Results section, Page 4 line 141 “… and one was a clinical trial (CT) [22]” also clinical trial (CT) has passed the inclusion criteria? Please explain in a more discursive way.

Response: In order to adapt to the reviewer's 2 comments, we have included in the Material and Method section that we also selected clinical trials which involved teeth in which endodontic access cavities were performed using computer‑aided techniques: either static or dynamic navigation techniques.

Reviewer 2: In Results section, Pag 4 line 143: Kindly add the reference of the mentioned study “and one study compared static and dynamic navigation with regard to conventional endodontic access”

Response: In order to adapt to the reviewer's 2 comments, we have added the reference.

Reviewer 2: In Discussion section, I suggest the following articles to enrich the introduction about this aspect: “Gambarini G, Galli M, Morese A, Stefanelli LV, Abduljabbar F, Giovarruscio M, Di Nardo D, Seracchiani M, Testarelli L. Precision of Dynamic Navigation to Perform Endodontic Ultraconservative Access Cavities: A Preliminary In Vitro Analysis. J Endod. 2020 Sep;46(9):1286-1290. doi: 10.1016/j.joen.2020.05.022. Epub 2020 Jun 15. PMID: 32553875.”

Response: In order to adapt to the reviewer's 2 comments, we have added the suggested reference.

Reviewer 2: In Discussion section, Page 9 line 225: kindly check the spelling “deviation (2.26° ± 1.62°)relative to” -> “deviation (2.26° ± 1.62°) relative to”

Response: In order to adapt to the reviewer's 2 comments, we have changed the sentence.

Reviewer 2: In Discussion section, Page 9 lines 231-232: I suggest adding reference [1] at this point. “dental malformations such as dens invaginatus [33,34] or dens evaginatus [35].”

Response: In order to adapt to the reviewer's 2 comments, we have not added the reference [1] because it was repeated with the reference [36].

Reviewer 2: Bibliography is not formatted respecting the journal’s requirements, no improper citations are evidenced. Bibliography is written in ascending order.

Response: In order to adapt to the reviewer's 2 comments, we have adapted the format of the references to the journal’s requirements.

We take this opportunity to thank the recommendations and suggestions made by the reviewers to improve the document.

Yours sincerely,

Reviewer 3 Report

An interesting analysis and a well-prepared study.

The reference section shows minor mistakes (double numbers, some mis-spellings)

Author Response

Dear Reviewer 3:

I’m pleased to resubmit the manuscript of the work entitled, “Efficacy of Computer-Aided Navigation Techniques on the Accuracy of Endodontic Access Cavities. A Systematic Review and Meta-Analysis”

Reviewer 3: The reference section shows minor mistakes (double numbers, some mis-spellings)

Response: In order to adapt to the reviewer's 3 comments, we have revised the references and corrected the mistakes.

We take this opportunity to thank the recommendations and suggestions made by the reviewers to improve the document.

Yours sincerely,